# Predicting Duration of Mechanical Ventilation in Acute Respiratory Distress Syndrome Using Supervised Machine Learning

**DOI:** 10.3390/jcm10173824

**Published:** 2021-08-26

**Authors:** Mohammed Sayed, David Riaño, Jesús Villar

**Affiliations:** 1Department of Computer Engineering, Universitat Rovira i Virgili, Av. Paisos Catalans 26, 43007 Tarragona, Spain; mgamal.sayed@urv.cat (M.S.); david.riano@urv.cat (D.R.); 2CIBER de Enfermedades Respiratorias, Instituto de Salud Carlos III, Monforte de Lemos 3-5, Pabellón 11, 28029 Madrid, Spain; 3Multidisciplinary Organ Dysfunction Evaluation Research Network, Research Unit, Hospital Universitario Dr. Negrín, Barranco de la Ballena s/n, 4th Floor-South Wing, 35019 Las Palmas de Gran Canaria, Spain; 4Keenan Research Center, Li Ka Shing Knowledge Institute, St. Michael’s Hospital, Unity Health Toronto, 38 Shuter St., Toronto, ON M5B 1A6, Canada

**Keywords:** intensive care unit, acute respiratory distress syndrome, mechanical ventilation, machine learning, prediction models

## Abstract

Background: Acute respiratory distress syndrome (ARDS) is an intense inflammatory process of the lungs. Most ARDS patients require mechanical ventilation (MV). Few studies have investigated the prediction of MV duration over time. We aimed at characterizing the best early scenario during the first two days in the intensive care unit (ICU) to predict MV duration after ARDS onset using supervised machine learning (ML) approaches. **Methods:** For model description, we extracted data from the first 3 ICU days after ARDS diagnosis from patients included in the publicly available MIMIC-III database. Disease progression was tracked along those 3 ICU days to assess lung severity according to Berlin criteria. Three robust supervised ML techniques were implemented using Python 3.7 (Light Gradient Boosting Machine (LightGBM); Random Forest (RF); and eXtreme Gradient Boosting (XGBoost)) for predicting MV duration. For external validation, we used the publicly available multicenter database eICU. **Results:** A total of 2466 and 5153 patients in MIMIC-III and eICU databases, respectively, received MV for >48 h. Median MV duration of extracted patients was 6.5 days (IQR 4.4–9.8 days) in MIMIC-III and 5.0 days (IQR 3.0–9.0 days) in eICU. LightGBM was the best model in predicting MV duration after ARDS onset in MIMIC-III with a root mean square error (RMSE) of 6.10–6.41 days, and it was externally validated in eICU with RMSE of 5.87–6.08 days. The best early prediction model was obtained with data captured in the 2nd day. **Conclusions:** Supervised ML can make early and accurate predictions of MV duration in ARDS after onset over time across ICUs. Supervised ML models might have important implications for optimizing ICU resource utilization and high acute cost reduction of MV.

## 1. Background

The acute respiratory distress syndrome (ARDS) is an important cause of morbidity, mortality, and costs in intensive care units (ICUs) worldwide [1]. It is a life-threatening form of acute respiratory failure characterized by inflammatory pulmonary edema leading to severe hypoxemia, requiring endotracheal intubation and mechanical ventilation (MV) in most cases [2]. The number of days on MV during the ICU stay is a major driver of high acute care costs [3,4,5]. We believe that an important intervention to mitigate these costs is timely recognition and treatment of conditions that can cause serious complications.

The Berlin definition of ARDS identifies three mutually exclusive categories of lung severity with PaO_2_/FiO_2_ ratios in the ranges >200–300 mmHg (mild ARDS), >100–200 mmHg (moderate ARDS), and ≤100 mmHg (severe ARDS) [6,7]. Some studies [8,9] have reported a progression of costs from mild, to moderate, to severe ARDS. Despite global acceptance of the Berlin criteria [10], some authors have questioned its ability to assess the “*true*” severity of lung injury [11]. A recent study argues that mild ARDS should be considered “severe in terms of level of care” [12]. This quality criterion (i.e., level of care) could be measured in terms of MV duration, but accurate predictions of MV duration are difficult for critical care physicians [13,14], particularly for patients requiring prolonged MV [14].

Predicting MV duration could influence important clinical decisions, such as timing of tracheostomy and initiation of oral nutrition [14]. In this context, one approach for an accurate prediction of MV duration is the use of artificial intelligence (AI) approaches, such as machine learning (ML). ML is a subset of AI in which machines extract knowledge from the data provided. ML is an exploratory process where there is no one-method-fits-all solution [15,16]. ML merges statistical analysis techniques with computer science to produce algorithms capable of “statistical learning” [17]. ML algorithms are divided into two categories: supervised and unsupervised [17]. Supervised learning algorithms, the ones used in our study, detect relationships between potential explanatory features and a known target outcome [16]. They are commonly used in ICUs to predict clinical outcomes [16,17,18,19,20,21]. Troché and Moine addressed the critical question on whether MV duration is predictable [22]. Herein, we present the use of three powerful supervised ML methods to develop novel models to predict MV duration in ARDS after onset over time, using the single-center MIMIC-III dataset under three different scenarios. Then, the eICU multicenter dataset was used to externally validate the best MIMIC-III prediction model.

## 2. Methods

### 2.1. Study Design and Patient Population

We used two publicly available clinical databases for development and external validation of the best ML predictive model: MIMIC- III [23] and eICU, respectively [24]. Data of the first 3 ICU days (day 1 for representative data within the first 24 h after ARDS onset, day 2 for data within 24–48 h after onset, and day 3 for data within 48–72 h after onset) (*n* = 2466, 1445, and 1278 patients, respectively) were extracted from the single-center dataset MIMIC-III (MetaVision, 2008–2012) [23]. Similarly, data of the first 3 ICU days after ARDS onset (*n* = 5153, 2981, and 2326 patients, respectively) were extracted from the multicenter dataset eICU (2014–2015) [24]. Patients <18 years were excluded. Data extraction from both datasets was performed using Python 3.7. The selection of clinical variables was based on prior studies [9,19,25,26,27]. All extracted patients from both datasets fulfilled the Berlin definition for ARDS [6]. For the purpose of this study, prolonged MV was defined as being ventilated for >48 h [22,28]. Disease progression in each dataset was tracked along those 3 ICU days.

### 2.2. MIMIC-III

Medical Information Mart for Intensive Care III (MIMIC-III) is a large single-center database containing de-identified health-related data of about 60,000 ICUs patients admitted to the Beth Israel Deaconess Medical Center (Boston, MA, USA) between 2001 and 2012 [23]. There were six predictors: baseline demographic information (age); ventilator parameters including PEEP; blood gas parameters including FiO_2_, PaO_2_, PaO_2_/FiO_2_, and PaCO_2_. The main target variable was MV duration.

### 2.3. eICU

eICU is a multicenter ICU database and it has a high granularity of data of more than 200,000 ICU admissions [24]. We used this database for external validation of the best prediction model obtained from MIMIC-III in order to obtain the MV duration prediction in the eICU database.

### 2.4. Predictive Models

During the first 24 h of ARDS onset, misdiagnosis can occur if clinicians consider qualifying PaO_2_ values resulting from acute events unrelated to the disease process (such as endotracheal tube obstruction, barotrauma, or hemodynamic instability), instead of considering only PaO_2_ values while patients are clinically stable. It is also well established that changes in PEEP and FiO_2_ within the first few hours of routine intensive care management alter the PaO_2_/FiO_2_ ratio in ARDS patients [11]. Since in a substantial proportion of patients diagnosed as having ARDS did not meet ARDS criteria within the first 24 h of care, we decided to examine supervised ML models in the following three scenarios during the first two ICU days: (i) scenario I: predicting MV duration using information captured in the 1st ICU day; (ii) scenario II: predicting MV duration using information captured in the 2nd ICU day; (iii) scenario III: predicting MV duration using information captured in the 1st and 2nd ICU days, then comparing these three scenarios with scenario IV for predicting MV duration using the information captured in the 3rd ICU day exclusively.

We implemented three robust supervised ML algorithms via Python 3.7, including Light Gradient Boosting Machine (LightGBM) [29], Random Forest (RF) [30], and eXtreme Gradient Boosting (XGBoost) [31] to generate predictive models for MV duration after ARDS onset over time in the development database. For external validation purposes, we used the multicenter eICU dataset, as these three methods sacrifice the explicitness of the model in favor of predictive quality, and the generated models should be seen as “black box” with a high predictive robustness. For the development database, we optimized each model’s parameters through a grid search over the respective model’s hyperparameter space and the quality of all prediction models was computed based on a 10-fold cross-validation approach, which means that the dataset was divided into 10 folds, and in each run, 9 were used for training, and the remaining 1 was used for testing. Root-mean-square error (RMSE) was used to assess the predictive quality of the models. RMSE flags more significant differences between the predicted and the actual patient readings when they occur [32]. MV duration was expressed in days.

## 3. Results

For development and validation databases, mean values and 95% confidence intervals (CI) of baseline parameters during the first three ICU days after ARDS onset are reported in Table 1. The median and interquartile range (IQR) of MV duration are reported in Table 2.

Table 3 shows the performance of the three supervised ML methods for the predictive scenarios in the development database. Table 4 shows the results of external validation of the best prediction model obtained from MIMIC-III to obtain the MV duration prediction in the eICU database.

For the development database, the best early ML model for predicting MV duration was obtained by scenario II with RMSE = 6.10 days, using LightGBM algorithm. Figure 1a represents the Bland–Altman plot for LightGBM prediction and truth values in scenario II.

For the validation database, the best early ML predictive model for MV duration was also observed for scenario II with RMSE = 5.87 days. This finding reinforces the idea that the best early approach for predicting MV duration is to consider the condition of the patient in the second ICU day after ARDS onset, rather than the first ICU day, or both. Figure 1b represents the Bland–Altman plot for prediction and truth values in scenario II using the external validation of LightGBM. The Bland–Altman plots illustrate agreement between the LightGBM models using the development and validation databases.

## 4. Discussion

Comparing the difference of RMSE means in the best early scenario (scenario II) with the prediction based on the data of patients in their third ICU day (scenario IV), yields minor RMSE differences (development database: 0.18 day (6.10–5.92) for LightGBM, and validation database: 0.16 day (5.87–5.71)). According to these low differences for both the development and validation datasets, our major finding was that the prediction results of LightGBM models based on the data of the second ICU day (scenario II) are very close to those corresponding results of LightGBM models based on the data of the third ICU day (scenario IV). Consequently, the LightGBM model can accurately predict MV duration without considering/waiting for the data of the third ICU day. This means that MV duration can be predicted earlier, and this could lead to better allocation of MV resources, reducing high acute costs of MV in ARDS, and improving patient care.

MV duration beyond 48 h in patients with ARDS provides information about risk factors in those patients [28] and has a direct correlation with ICU costs [4,5]. An early predictive model for MV duration can optimize ICU-level resource utilization [5,33]. Previous attempts to predict MV duration using conventional ICU scores or traditional statistical regression based techniques have proven to be difficult and failed to deal with the diversity of big data in the modern ICU databases [22]. ML is reliable, and it is a non-invasive modality to generate models for predicting MV duration. Most previous works considered a discriminatory prediction model to determine if a patient will remain intubated after a fixed number of days (e.g., 7 days) [22]. By contrast, our approach is numerical, and it predicts the number of MV days earlier by using commonly accessible clinical variables during the first two ICU days. Furthermore, to strengthen the evidence of our results, we used a multicenter database (eICU) for external validation, in which the best model obtained from a single-center database (MIMIC-III) was used to obtain the MV duration prediction in the eICU database. Our findings could be used to facilitate optimal triage, more timely management, and ICU resource utilization [34]. They may also affect some important clinical decisions, including timing of tracheostomy and, potentially, transfers to long-term ventilator weaning units or referral to other centers [13].

Herein, the main objective of using ML was to show that the application of ML is a promising approach to predict MV duration early. The ML contribution in this large study is to demonstrate the applicability of this approach, while not trying to choose the most proper ML model. Furthermore, we believe that the results of an efficient ML technique can yield accurate results for predicting MV duration. In terms of clinical relevance, our ML findings showed that using clinical data from the first ICU day is less predictive than data from the second ICU day. Previous studies showed that the accuracy of intensivists to predict MV duration is limited [13]. However, comparison to other published ML prediction of MV duration is difficult, as we aimed at predicting MV duration for MV >48 h and prior studies predicted for different outcomes under different time frames, in different populations, and using different ML metrics. A recent ML study showed that RMSE for predicting MV duration in ARDS patients for MV >48 h, was 6.23 days [9]. However, this study in [9] had several weaknesses: (1) it ignored the temporal dependency of the longitudinal predictor and treated each observed data point independently, and (2) it was only based on the single-center MIMIC-III database without external validation. Hence, those findings have serious limitations for the generalizability in the context of assessing the prediction of ARDS outcome.

From the cost perspective, the mean incremental cost of MV in ICU patients in the US was $1522 per day [4]. For instance, if we compare our findings with the result of the best ML method used in [9], which had a RMSE of 6.23 days, we see that LightGBM approach (the best approach) improved the current state of the art. This improvement can be quantified in terms 0.13 day (6.23–6.10) and about US $198 per patient according to [4]. Developing early predictive models using ML could assist to implement policies for the reduction of high acute care costs in ARDS [3,4,5]. Previous clinical studies showed acute costs incurred by mechanically ventilated ICU patients, but there is a significant difference in costs between ventilated ARDS patients and those without ARDS [35]. More specifically, ARDS diagnosis increases total ICU and hospital costs for mechanically ventilated ICU patients, suggesting higher total costs due to more days on a ventilator, although there is no clear severity-dependent relationship between ARDS severity and incurred costs [35]. The benchmarking of ML algorithms is possible through publicly available databases such as MIMIC-III [19,27] or eICU [19,36].

We acknowledge that our study has several strengths. First, we have analyzed a large population of over 7000 ARDS patients from two ICU databases within the first three ICU days after ARDS onset. Second, we have implemented and externally validated the best ML model (LightGBM) that can predict MV duration early and accurately using commonly accessible clinical variables. Third, early prediction of MV duration can inform population-level ICU resource allocation. Despite its strengths, we also acknowledge some limitations. First, our study is based on a retrospective analysis of data and should be confirmed through further prospective studies. Second, one could argue that the outcome of MV duration is somewhat subjective and could be a function of local practice or intrinsic bias inherent in such critical care decisions. However, our ability to predict a clinically relevant and difficult-to-predict outcome (MV duration) early supports the value of the proposed supervised ML models.

## 5. Conclusions

Predicting MV duration after ARDS onset over time is complex and cannot be adequately performed by critical care physicians. Our findings showed that the ML-based early prediction of MV duration is more accurate when predictive models are based on the clinical features of ARDS patients in the second ICU day after ARDS onset.

## Figures and Tables

**Figure 1 jcm-10-03824-f001:**
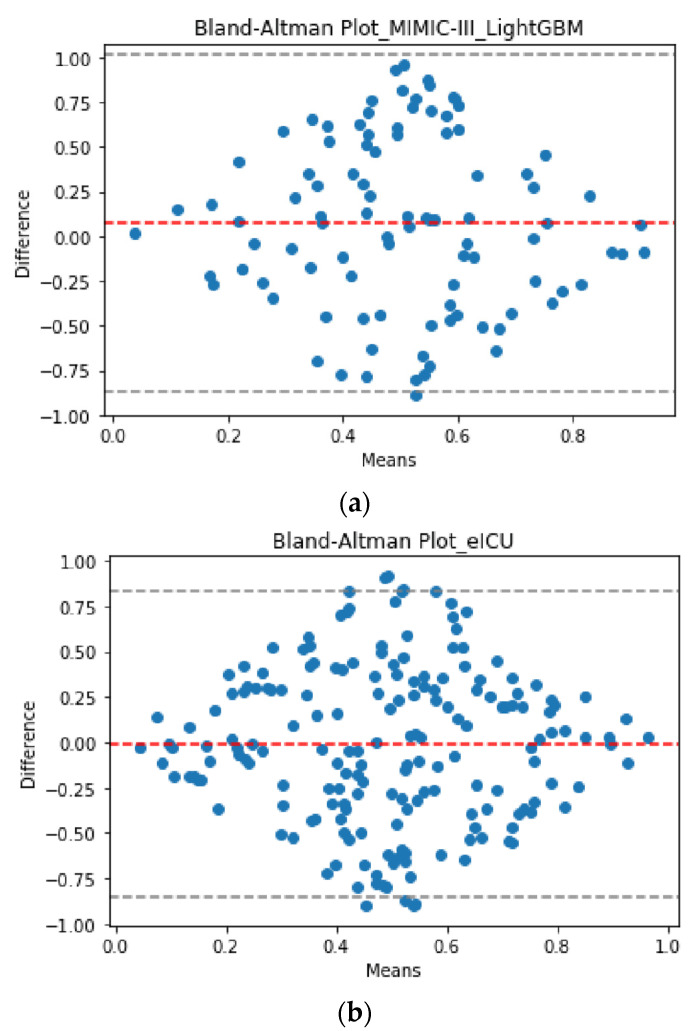
Bland–Altman plot for the truth vs. the predicted values of MV duration using LightGBM (the best validated model) in Scenario II (the best early scenario). (**a**) Development database; (**b**) validation database. The X- and Y-axes stand for the mean and the difference of the two measurements, respectively. Please note that the values shown in the Bland–Altman plot are normalized in the interval (0, 1) (i.e., values are scaled to have corresponding values between 0 and 1).

**Table 1 jcm-10-03824-t001:** Predictors and their descriptive statistics in MIMIC-III and eICU at 24 h, 48 h, and 72 h.

	24-h	48-h	72-h
*A. MIMIC-III ARDS Patients*	*2466 (100%)*	*1445 (58.6%)*	*1278 (51.8%)*
*B*. *Means and 95% CI* Age	62.2 (61.5, 62.8]	60.8 (59.9, 61.6)	60.9 (60.0, 61.8)
PEEP	7.6 (7.5, 7.7)	9.1 (8.9, 9.4)	8.9 (8.8, 9.2)
FiO_2_	0.66 (0.65, 0.67)	0.54 (0.53, 0.55)	0.51 (0.49, 0.51)
PaO_2_	114.5 (112.8, 116.2)	97.6 (96.3, 98.9)	95.4 (94.1, 96.6)
PaCO_2_	43.4 (42.9, 43.9)	42.3 (41.8, 42.9)	42.9 (42.4, 43.6)
PaO_2_/FiO_2_	184.3 (181.9, 186.6)	170.9 (167.7, 174.2)	179.1 (175.7, 182.5)
*C. eICU ARDS Patients*	*5153 (100%)*	*2981 (57.8%)*	*2326 (45.1%)*
*D*. *Means and 95% CI* Age	63.4 (62.9, 63.8]	63.4 (62.8, 63.9)	62.9 (62.4, 63.6)
PEEP	6.6 (6.6, 6.7)	7.1 (7.0, 7.2)	7.3 (7.1, 7.4)
FiO_2_	0.63 (0.63, 0.64)	0.53 (0.52, 0.54)	0.52 (0.51, 0.53)
PaO_2_	104.1 (102.9, 105.2)	89.1 (88.1, 90.1)	86.4 (85.3, 87.4)
PaCO_2_	43.5 (43.2, 43.9)	41.3 (40.9, 41.7)	41.8 (41.4, 42.2)
PaO_2_/FiO_2_	160.2 (158.3, 162.1)	175.2 (172.9, 177.5)	174.5(171.8, 177.2)

**Table 2 jcm-10-03824-t002:** MV Duration in ARDS across MIMIC-III and eICU.

ICU Day (*n*)	Database	MV Duration *Median Days (IQR Days)*
Day 1 (*2466*)	*MIMIC-III*	6.5 (4.4–9.8)
Day 2 (*1445*)		6.8 (4.7–10.5)
Day 3 (*1278*)		6.9 (4.7–10.6)
Day 1 (5153)	*eICU*	5.0 (3.0–9.0)
Day 2 (2981)		6.0 (4.0–10.0)
Day 3 (2326)		6.0 (4.0–10.0)

**Table 3 jcm-10-03824-t003:** Performances of LightGBM, RF, and XGBoost models to predict MV duration over time in MIMIC-III.

Scenario I: Predicting MV Duration in ARDS Using Data in the 1st ICU Day	
**Algorithm**	**RMSE, mean** *±* **SD**
*XGBoost*	*6.81 ± 1.18*
*RF*	*6.79 ± 1.22*
*LightGBM*	***6.41** ± **1.55***
***** Scenario II: Predicting MV duration in ARDS using data in the 2nd ICU day**	
**Algorithm**	**RMSE, mean** *±* **SD**
*XGBoost*	*6.53 ± 0.96*
*RF*	*6.55 ± 1.16*
** **LightGBM***	***6.10** ± **0.72***
**Scenario III: Predicting MV duration in ARDS using data in the 1st & 2nd ICU days**	
**Algorithm**	**RMSE, mean** *±* **SD**
*XGBoost*	*6.57 ± 1.08*
*RF*	*6.60 ± 1.01*
***LightGBM***	***6.35** ± **0.69***
**Scenario IV: Predicting MV duration in ARDS using the data in the 3rd ICU day**	
**Algorithm**	**RMSE, mean** *±* **SD**
*XGBoost*	*6.14 ± 0.85*
*RF*	*6.19 ± 0.66*
***LightGBM***	***5.92** ± **0.47***

***** Identifies the optimal scenario and ML model.

**Table 4 jcm-10-03824-t004:** External validation of the best prediction model (LightGBM) obtained from MIMIC-III to obtain the MV duration prediction in the eICU database.

Predictive Scenario	RMSE, Mean *±* SD
*Scenario I*	*6.08 ± 0.72*
** **Scenario II***	***5.87** ± **0.67***
*Scenario III*	*5.93 ± 0.44*
*Scenario IV*	*5.71 ± 0.55*

***** Identifies the optimal scenario and ML model.

## Data Availability

By reasonable request to M.S. and D.R.

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
