# Peer review of "Predicting Duration of Mechanical Ventilation in Acute Respiratory Distress Syndrome Using Supervised Machine Learning"

_jcm, 2021, doi:10.3390/jcm10173824_

Round 1

Reviewer 1 Report

GENERAL COMMENTS:

Using publicly available data, the authors retrospectively generate a machine learning model for prediction of mechanical ventilation duration in ARDS patients during the first 3 days after ICU ARDS diagnosis .  A strength of this study is the use of 2 independent ICU databases, one to generate the prediction model and the other used to externally validate the model.  The manuscript is difficult to read and likely to be so for readers of this technical statistical paper, when published in a clinical journal.  I think the authors can clarify their report by restricting the results and discussion to the *LightGBM results of 6.10 ± 0.72  RMS error days.

This would enable the authors to consolidate a number of tables and focus on their "best method."  I think the readers will be distracted by reporting the other 3 statistical methods (scenarios) and do not believe they add useful information.  The authors could then expend more effort explaining their "best result" and clarifying why this might be important.  The manuscript can be clarified by emphasizing the RMS Error results, labeling the results as error in days, and linking the error results to a new, currently missing, distribution of predicted mechanical ventilation durations in days.  That should enable the clinician reader to interpret this report.  Improving the display and explanation of the Bland-Altman plots would help achieve this goal.

SPECIFIC COMMENTS:

P 1 Lines 42-43: "

 important intervention to mitigate these costs is timely recognition and treatment of conditions that can cause serious complications" is a declarative statement that sounds like fact but is without supporting evidence.  The authors should consider modifying this with "We believe" or "It is reasonable to expect" or some such indication of their opinion.

P 2 Lines 87-88: "predictors were: baseline demographic information (age); ventilator parameters including PEEP; blood gas parameters including FiO2, PaO2, PaO2/FiO2, 88 and PaCO2. " - I suggest the authors make these descriptions in a consistent style.  Are the demographic parameters only age or do they include age?

P 3 Lines 93-4: 'in order to obtain the MV duration prediction 93 in the eICU database. ' is not clear.  Was the eICU database used to validate the best MIMIC prediction model, or did the eICU validation lead to a new or revised prediction model?  Please clarify.

Lines 96-7: 'a substantial proportion of patients diagnosed as having ARDS did not meet 96 ARDS criteria within the first 24 hours of care ' is disturbing.  This suggests the quality of the information in the MIMIC database is lower than desired.  It raises an important question about replicability, a fundamental requirement of scientific studies.  The absence of replicability in medical and other studies is currently viewed as a crisis in science by many.  this deserves a comment regarding the quality of the data in the database.

Lines 107-8: "sacrifice the explicitness of the model in favor 107 of predictive quality, so that the generated models should be seen as “black box " leads to results that may be difficult for clinicians to accept because they cannot assess the method.  Acceptance, then depends on credible evidence that the model works well in clinical practice when used for decision-making.  I suggest the authors consider mentioning this as a reason for selecting a second, independent database for validation.  This is a strength of their study design.

Lines 109-110: "optimized each model’s parameters through a grid search 109 over the respective model’s hyperparameter space " raises, again an fundamental question about their method.  Did they sue the eICU database to validate the MIMIC database model, or did they use the eICU database to modify and improve the model?  Please clarify.

Line 111: "10-fold cross-validation approach " is not clear to this reviewer.  Please explain what this means.  I suspect it is linked to my questions about validation only or modification of the MIMIC model with the eICU database.

Lines 112-3: "RMSE quantifies more 112 significant differences " will not be easily understood by many readers.  Please clarify what this means.

Line 117: "baseline parameters (age, PEEP, gas-exchange) " are these the only parameters or are these examples?  If examples, precede the list with "e.g.," - After viewing Table 1 it appears simpler to just remove the parenthetical and its contents from this sentence.

Line 120: Table 1. Predictors " it seems that only 6 predictors were utilized and listed in Table 1.  I suggest the authors make this clear in their preceding text.  What does the % following the patient number at the top of the table mean?  Please explain.

Line 124: "Table 3 shows the quality of the three supervised ML methods " would be better as "Table 3 contains the root mean square errors ≠Sd in days for prediction of Mechanical Ventilation in the MIMIC database."  If I understand correctly, you are presenting RMS errors in prediction and not duration of mechanical ventilation.  This is not clear but must be clearly explained to enable readers to understand the study.  The use of the word "quality" is, in my view, inappropriate.  You are presenting errors. I have similar comments and suggestions for Table 4 (line 131).

Lines 134-5: "best early ML model for predicting MV duration 134 was obtained by scenario II with RMSE = 6.10 days, using LightGBM algorithm " reporting the best early ML model does not indicate if the model is good enough at predicting mechanical ventilation duration to be clinically useful.  What I believe is missing and could enhance the value of this report is a compelling graphic link between the RMS Error in days to the distribution of predicted mechanical ventilation in days.  Figure 1a and b, the Bland-Altman plot, are not satisfactory in my view.  The differences are not labeled, the means are not labeled, and I cannot easily link the RMS Errors in days to predicted days of mechanical ventilation.

Lines 137-9: "values shown in the Bland-Altman plot are normalized in the interval [0,1]. Differ-137 ences between predicted and actual values showed a relatively spread around 0 across the 138 range of MV duration average ' is not at all clear.  Please explain in plain English.

Lines 142-3: "best early approach for predicting MV duration is to consider the condition of 142 the patient in the second ICU day after ARDS " may be true but does not address the potential clinical significance of their result.

Lines 174-7:  "Our findings could be used to facilitate optimal 174 triage, more timely management, and ICU resource utilization [34]. They may also affect 175 some important clinical decisions, including timing of tracheostomy and potentially trans-176 fers to long-term ventilator weaning units or referral to other centers "  this will only be true if the results have clinical validity.  The authors will strengthen their report if they pursue the link between RMS error days and predicted days of mechanical ventilation mention several times in preceding comments.

Lines 178-9: "application of ML is a 178 promising approach to early predict MV duration "  The authors have not yet shown this is clinically promising.

Lines 195-6 and following: "From the cost perspective, the mean incremental cost of MV in ICU patients in the 195 US was $1,522 per day "  I think the discussion of cost is not pertinent to this report.  Cost is, of course, interesting but neither the subject of, not central to central to ,their report and study.

Lines 222-4:  "In summary, the application of predictive models generated by supervised ML meth-222 odds is a promising research area to optimize resource utilization and reduce acute costs in 223 ICUs. " is not pertinent to the study and result reported in this manuscript.  I would remove this from the conclusion.

Lines 227-32: "Our ML 227 results reinforced the idea that data of patients in their second ICU day is more predictive 228 for MV duration than data from the first ICU day after ARDS onset. Our supervised ML 229 algorithms aimed at determining the potential clinical utility of the proposed early pre-230 diction model for optimizing timing of tracheal intubation, better allocation of MV re-231 sources and staff, reducing high acute care costs in ARDS, and improving patient care. " should be omitted.  The first sentence is redundant, and the second sentence recapitulates the introduction and is not a conclusion derived from this study.

Author Response

R0- Thank you very much for your positive comments about our study.

Following your recommendation, we have modified the manuscript to make it easier to understand by the readers.

Q1) P 1 Lines 42-43: "important intervention to mitigate these costs is timely recognition and treatment of conditions that can cause serious complications" is a declarative statement that sounds like fact but is without supporting evidence.  The authors should consider modifying this with "We believe" or "It is reasonable to expect" or some such indication of their opinion.

R1- Thank you. Following your recommendation, we have added “We believe that…” at the beginning of this statement.

Q2) P 2 Lines 87-88: "predictors were: baseline demographic information (age); ventilator parameters including PEEP; blood gas parameters including FiO2, PaO2, PaO2/FiO2, 88 and PaCO2." - I suggest the authors make these descriptions in a consistent style.  Are the demographic parameters only age or do they include age?

R2- Thank you for this observation. We have modified the sentence by stating “The predictors were six: …” 

Q3) P 3 Lines 93-4: 'in order to obtain the MV duration prediction in the eICU database. 'is not clear. Was the eICU database used to validate the best MIMIC prediction model, or did the eICU validation lead to a new or revised prediction model?  Please clarify.

R3- Thank you. We think that the current statement “We used this database for external validation of the best prediction model obtained from MIMIC-III in order to obtain the MV duration prediction in the eICU database” is clear enough for stating that the eICU database was used to externally validate the best MIMIC prediction model.

Q4) Lines 96-7: 'a substantial proportion of patients diagnosed as having ARDS did not meet ARDS criteria within the first 24 hours of care ' is disturbing.  This suggests the quality of the information in the MIMIC database is lower than desired.  It raises an important question about replicability, a fundamental requirement of scientific studies.  The absence of replicability in medical and other studies is currently viewed as a crisis in science by many.  This deserves a comment regarding the quality of the data in the database.

R4- Thank you for this comment. In the revised manuscript, we have now added “During the first 24 hours of ARDS onset, misdiagnosis can occur if clinicians consider qualifying PaO2 values resulting from acute events unrelated to the disease process (such as endotracheal tube obstruction, barotrauma, or hemodynamic instability), instead of considering only PaO2 values while patients are clinically stable. It is also well established that changes in PEEP and FiO2 within the first few hours of intensive care management alter the PaO2/FiO2 in ARDS patients [11].”

Q5) Lines 107-8: "sacrifice the explicitness of the model in favor of predictive quality, so that the generated models should be seen as “black box " leads to results that may be difficult for clinicians to accept because they cannot assess the method.  Acceptance, then depends on credible evidence that the model works well in clinical practice when used for decision-making.  I suggest the authors consider mentioning this as a reason for selecting a second, independent database for validation.  This is a strength of their study design.

R5- Thank you! As requested, in the revised manuscript we have now modified this paragraph by stating: “We implemented three robust supervised ML algorithms via Python 3.7, including Light Gradient Boosting Machine (LightGBM) [29], Random Forest (RF) [30], and eXtreme Gradient Boosting (XGBoost) [31] to generate predictive models for MV duration after ARDS onset over time in the development database. For external validation purposes, we used the multicenter eICU dataset as these three methods sacrifice the explicitness of the model in favor of predictive quality, and the generated models should be seen as “black box” with a high predictive robustness. For the development database, we optimized each model’s parameters through a grid search over the respective model’s hyperparameter space and the quality of all prediction models was computed based on a 10-fold cross-validation approach, which means that the dataset was divided into 10 folds, and in each run, 9 were used for training and the remaining 1 was used for testing. Root-mean-square error (RMSE) was used to assess the predictive quality of the models. RMSE flags more significant differences between the predicted and the actual patient readings when they occur [32].”

Q6) Lines 109-110: "optimized each model’s parameters through a grid search over the respective model’s hyperparameter space" raises, again a fundamental question about their method.  Did they sue the eICU database to validate the MIMIC database model, or did they use the eICU database to modify and improve the model?  Please clarify.

R6- Done! Please, see R5.

Q7) Line 111: "10-fold cross-validation approach" is not clear to this reviewer.  Please explain what this means.  I suspect it is linked to my questions about validation only or modification of the MIMIC model with the eICU database

R7- Done! Please, see R5.

Q8) Lines 112-3: "RMSE quantifies more significant differences" will not be easily understood by many readers.  Please clarify what this means

R8- Done! Please, see R5.

Q9) Line 117: "baseline parameters (age, PEEP, gas-exchange)" are these the only parameters or are these examples?  If examples, precede the list with "e.g.," - After viewing Table 1 it appears simpler to just remove the parenthetical and its contents from this sentence.

R9- Thank you. As requested, we have removed the parenthesis and its contents from this sentence.

Q10) Line 120: Table 1. Predictors "it seems that only 6 predictors were utilized and listed in Table 1.  I suggest the authors make this clear in their preceding text.  What does the % following the patient number at the top of the table mean?  Please explain.

R10- Thank you. Please, also see R2. The sign “%” following the patient number at the top of the table represents the percentage of patients in the second and third ICU days relative to the total patients in the first ICU day. For instance, in MIMIC-III the total number of patients in the first ICU day is 2466 which is 100%, so the percentage of patients in the second and third ICU days relative to the total patients in the first ICU day equals 1445/2466 or 58.6% and 1278/2466 or 51.8%, respectively.

Q11) Line 124: "Table 3 shows the quality of the three supervised ML methods" would be better as "Table 3 contains the root mean square errors ≠Sd in days for prediction of Mechanical Ventilation in the MIMIC database."  If I understand correctly, you are presenting RMS errors in prediction and not duration of mechanical ventilation.  This is not clear but must be clearly explained to enable readers to understand the study.  The use of the word "quality" is, in my view, inappropriate.  You are presenting errors. I have similar comments and suggestions for Table 4 (line 131).

R11- Sorry. In the revised manuscript, we have changed the sentence as: “Table 3 shows the performance of the three supervised ML methods for the predictive scenarios in the development database”. For Table 4 (line 131), we think that there is no need for more clarification.

Q12) Lines 134-5: "best early ML model for predicting MV duration was obtained by scenario II with RMSE = 6.10 days, using LightGBM algorithm " reporting the best early ML model does not indicate if the model is good enough at predicting mechanical ventilation duration to be clinically useful.  What I believe is missing and could enhance the value of this report is a compelling graphic link between the RMS Error in days to the distribution of predicted mechanical ventilation in days.  Figure 1a and b, the Bland-Altman plot, are not satisfactory in my view.  The differences are not labeled, the means are not labeled, and I cannot easily link the RMS Errors in days to predicted days of mechanical ventilation.

R12- Please, see R16. Usually, in the Bland-Altman plot, X- and Y-axes stand for the mean and the difference of the two measurements, respectively, as reported in our manuscript. Please, see Figure 1 legend. Regarding your suggestion for the graph, please, see R15.

Q13) Lines 137-9: "values shown in the Bland-Altman plot are normalized in the interval [0,1]. Differences between predicted and actual values showed a relatively spread around 0 across the range of MV duration average ' is not at all clear.  Please explain in plain English.

R13- Thank you for this comment. Lines 137-139: “The values shown in the Bland-Altman plot are normalized in the interval [0,1]. Differences between predicted and actual values showed a relatively spread around 0 across the range of MV duration average.” and lines 145-147, the sentences “The values shown in the Bland-Altman plot are normalized in the interval [0,1]. Differences between predicted and actual values showed a relatively spread around 0 across the range of MV duration average” have been deleted in the revised manuscript.

We have now added in the revised manuscript a new interpretation after line 144 as a separate paragraph, stating: “The Bland-Altman plots illustrating agreement between the LightGBM models using the development and validation databases.”

Q14) Lines 142-3: "best early approach for predicting MV duration is to consider the condition of the patient in the second ICU day after ARDS" may be true but does not address the potential clinical significance of their result.

R14- Thanks for this observation. We aim to characterize the best early scenario to predict MV duration after ARDS onset over time using machine learning approaches. Please, see our response R16.

Q15) Lines 174-7:  "Our findings could be used to facilitate optimal triage, more timely management, and ICU resource utilization [34]. They may also affect some important clinical decisions, including timing of tracheostomy and potentially transfers to long-term ventilator weaning units or referral to other centers" this will only be true if the results have clinical validity.  The authors will strengthen their report if they pursue the link between RMS error days and predicted days of mechanical ventilation mention several times in preceding comments

R15- If we understand you correctly, your line of thought about “the link between RMSE error days and predicted days of mechanical ventilation”, we may say that the idea is inapplicable, as it merges between two different concepts of “Prediction vs. Forecasting”. In supervised machine learning, we are only concerned with predictions. Prediction is concerned with estimating the outcomes for unseen data. For this purpose, you must fit a model to a training dataset, which results in an estimator  (x) that can make predictions for new samples x.

Q16) Lines 178-9: "application of ML is a promising approach to early predict MV duration" The authors have not yet shown this is clinically promising.

R16- For both development and validation datasets, we demonstrated that the prediction results of LightGBM models based on the data of second ICU day (the best early prediction scenario, which is scenario II) are very close to those corresponding results of LightGBM models based on the data of the third ICU day (scenario IV). Therefore, we concluded that LightGBM model can accurately predict MV duration without considering/waiting for the data of the third ICU day. This conclusion means that MV duration can be predicted earlier, and this will lead to better allocation of MV resources, reducing high acute MV costs in ARDS, and improving patient care.

We have added a new paragraph for clarifying that in the discussion section of the revised manuscript.

Q17) Lines 195-6 and following: "From the cost perspective, the mean incremental cost of MV in ICU patients in the US was $1,522 per day" I think the discussion of cost is not pertinent to this report.  Cost is, of course, interesting but neither the subject of, not central to central to their report and study.

R17- We agree with the Reviewer that the aspect of cost is not central to the analysis of this study. However, the number of days on MV during the ICU stay is a major driver of high acute care costs [3-5]. Previous clinical studies showed acute costs incurred by mechanically ventilated ICU patients, but there is a significant difference in costs between ventilated ARDS patients and those without ARDS [35]. More specifically, ARDS diagnosis increases total ICU and hospital costs for mechanically ventilated patients, suggesting higher total costs due to more days on a ventilator, although there is no clear severity-dependent relationship between ARDS severity and incurred costs [35].

We would like to ask to the Reviewer to maintain this very short discussion on the financial cost perspective, as it could have an impact on improving patient care.

Q18) Lines 222-4:  "In summary, the application of predictive models generated by supervised ML methods is a promising research area to optimize resource utilization and reduce acute costs in ICUs." is not pertinent to the study and result reported in this manuscript.  I would remove this from the conclusion.

R18- Thank you for this comment. This sentence has been deleted in the revised manuscript.

Q19) Lines 227-32: "Our ML 227 results reinforced the idea that data of patients in their second ICU day is more predictive for MV duration than data from the first ICU day after ARDS onset. Our supervised ML algorithms aimed at determining the potential clinical utility of the proposed early prediction model for optimizing timing of tracheal intubation, better allocation of MV resources and staff, reducing high acute care costs in ARDS, and improving patient care” should be omitted.  The first sentence is redundant, and the second sentence recapitulates the introduction and is not a conclusion derived from this study.

R19- Thank you for this comment. Those sentences have been deleted in the revised manuscript.

Reviewer 2 Report

Dear authors,

I reviewed your manuscript titled " Prediciting duration of mechanical ventilation in the acute respiratory distress syndrome using supervised machine learning" and I found it really interesting. In my point of view it will be appropiate to describe which parameters/factors are the different algorithms used to calculate the mechanical ventilation duration. Which parameters are included in the databases for the posterior analysis (age, PaO2/Fio2, scores?).

In my opinion is also missing, into the discussion section, why knowing the mechanical ventilation duration is important at early stages of the ICU admission, how does this improve the clinical practice or the ICU/hospital organization?

Author Response

Q0) I reviewed your manuscript titled "Predicting duration of mechanical ventilation in the acute respiratory distress syndrome using supervised machine learning" and I found it really interesting.

R0- We thank the Reviewer for the positive comments about our study.

Q1) In my point of view it will be appropriate to describe which parameters/factors are the different algorithms used to calculate the mechanical ventilation duration. Which parameters are included in the databases for the posterior analysis (age, PaO2/Fio2, scores?).

Q2) In my opinion is also missing, into the discussion section, why knowing the mechanical ventilation duration is important at early stages of the ICU admission, how does this improve the clinical practice or the ICU/hospital organization?

R2- Thank you for the opportunity to clarify these issues.

For both development and validation datasets, our findings showed that the prediction results of LightGBM models based on the data of second ICU day (the best early prediction scenario, which is scenario II) are very close to those corresponding results of LightGBM models based on the data of the third ICU day (scenario IV).

Therefore, we concluded that LightGBM model can accurately predict MV duration without considering/waiting the data of the third ICU day. This means that MV duration can be predicted earlier, and it will lead to better allocation of MV resources, reducing high acute MV costs in ARDS, and improving patient care. ML approaches can accurately predict MV duration in ARDS after onset and would support clinical decisions that could improve patient outcome in ARDS patients.

Following your recommendation, we have added a new paragraph for clarifying that in the discussion section of the revised manuscript.
